# Genetic deletion of mast cell serotonin synthesis prevents the development of obesity and insulin resistance

Julian M. Yabut[1,2], Eric M. Desjardins[1,2], Eric J. Chan[1,3], Emily A. Day[1,2], Julie M. Leroux[1,2], Bo Wang [1,2], Elizabeth D. Crane[4], Wesley Wong [4], Katherine M. Morrison [1,5], Justin D. Crane [4], Waliul I. Khan[1,6,7] & Gregory R. Steinberg [1,2,3]*

Obesity is linked with insulin resistance and is characterized by excessive accumulation of adipose tissue due to chronic energy imbalance. Increasing thermogenic brown and beige adipose tissue futile cycling may be an important strategy to increase energy expenditure in obesity, however, brown adipose tissue metabolic activity is lower with obesity. Herein, we report that the exposure of mice to thermoneutrality promotes the infiltration of white adipose tissue with mast cells that are highly enriched with tryptophan hydroxylase 1 (Tph1), the rate limiting enzyme regulating peripheral serotonin synthesis. Engraftment of mast cell-deficient mice with Tph1$^{-/-}$ mast cells or selective mast cell deletion of Tph1 enhances uncoupling protein 1 (Ucp1) expression in white adipose tissue and protects mice from developing obesity and insulin resistance. These data suggest that therapies aimed at inhibiting mast cell Tph1 may represent a therapeutic approach for the treatment of obesity and type 2 diabetes.

[1] Centre for Metabolism, Obesity and Diabetes Research, McMaster University, 1280 Main St. W., Hamilton, ON, Canada L8N 3Z5. [2] Division of Endocrinology and Metabolism, Department of Medicine, McMaster University, 1280 Main St. W., Hamilton, ON, Canada L8N 3Z5. [3] Department of Biochemistry and Biomedical Sciences, McMaster University, 1280 Main St. W., Hamilton, ON, Canada L8N 3Z5. [4] Department of Biology, Northeastern University, Boston, MA 02115, USA. [5] Department of Pediatrics, McMaster University, 1280 Main St. W, Hamilton, ON, Canada L8N 3Z5. [6] Farncombe Family Digestive Health Research Institute, McMaster University, 1280 Main St. W., Hamilton, ON, Canada L8N 3Z5. [7] Department of Pathology and Molecular Medicine, McMaster University, 1280 Main St. W., Hamilton, ON, Canada L8N 3Z5. *email: gsteinberg@mcmaster.ca

O besity is tightly linked with the development of non-alcoholic fatty liver disease (NAFLD), insulin resistance and cardiovascular disease and is characterized by an excessive accumulation of adipose tissue caused by an imbalance of energy intake and expenditure[1,2]. Brown adipose tissue (BAT) has a high capacity for fatty acid and glucose oxidation due to the presence of futile cycling mediated by mitochondrial uncoupling protein 1 (Ucp1)[3,4], which is activated by β-adrenergic stimuli in response to cold or nutrients[5]. Ucp1[+] beige/brite adipocytes are also interspersed within white adipose tissue (WAT) and can be recruited for non-shivering thermogenesis[3,4,6]. Since rodent beige/brite adipocytes have a molecular signature reminiscent of supraclavicular BAT in lean adult humans[7–9], studies aimed at understanding beige adipose tissue in rodents may have direct implications for human BAT. Human BAT can be stimulated by cold or β-adrenergic stimuli as measured by an increase in [18]F-labeled fluorodeoxyglucose (FDG) uptake measured with positron emission tomography/computerized tomography (PET/CT)[10,11]. However, the ability of cold to increase BAT metabolic activity is impaired in the obese compared to lean healthy controls, despite greater BAT volume[12]. In both rodents and humans, norepinephrine is a critical hormone required for the maintenance of adipose tissue thermogenic capacity;[3,4] however, the signals that inhibit this pathway during obesity remain incompletely understood[12].

A potentially important inhibitor of adipose tissue thermogenesis is serotonin[13,14]. Previous studies have indicated that genetic and chemical inhibition of the rate limiting enzyme regulating peripheral serotonin synthesis, tryptophan hydroxylase 1 (Tph1), enhances adipose tissue thermogenesis and protects mice against obesity and insulin resistance[13,14]. Circulating serotonin levels in the periphery are dependent on the expression of Tph1 within the gastrointestinal tract (for review see refs. [15,16]), yet surprisingly, the genetic deletion of Tph1 within the gastrointestinal tract does not result in a lean phenotype[17], which is in contrast to the anti-obesity phenotype of germline deletion[13,14]. While a previous study using Tph1 floxed mice crossed to mice expressing Ap2-Cre suggested adipocyte Tph1 was important for inhibiting adipose tissue thermogenesis[13], it is known that the Ap2-Cre is expressed in many different immune cells found within the stromal vascular fraction[18]. Thus, the primary Tph1 expressing cell type(s) that inhibit adipose tissue thermogenesis are currently unknown.

Here, we find that mice housed at thermoneutrality have elevations in mast cells, Tph1 and serotonin in WAT that are associated with reductions in Ucp1. In mice fed a high-fat diet, genetic removal of Tph1 in mast cells from two distinct mouse models elevates basal metabolic rate, increases WAT Ucp1 and protects mice from obesity, insulin resistance and fatty liver disease compared to relevant controls. These data establish a role for mast cells in regulating adipose tissue thermogenesis and suggest that the therapeutic targeting of mast cell Tph1 may be a future strategy for the treatment of obesity and related metabolic disorders including insulin resistance and NAFLD.

## Results

### Thermoneutrality increases WAT Tph1 in HFD-fed mice. To delineate the potential role of Tph1 and peripheral serotonin for inhibiting adipose tissue thermogenesis and the primary cell type(s) that might be mediating this effect, we first conducted experiments in mice housed at thermoneutrality (TN; 29 °C); a condition known to dramatically reduce adipose tissue thermogenesis compared to housing mice at room temperature (RT; 22 °C)[19,20]. Mice housed at thermoneutrality had reductions in oxygen consumption (Supplementary Fig. 1a), energy

expenditure (Supplementary Fig. 1b) and BAT activity (Supplementary Fig. 1c, d), effects which were independent of changes in body mass (Supplementary Fig. 1e) or fat mass (Supplementary Fig. 1f). As anticipated, thermoneutral housing reduced Ucp1 expression in all adipose tissue depots (Fig. 1a). We subsequently examined Tph1 expression and found that it was unchanged in BAT, but was significantly elevated in inguinal WAT (iWAT) and gonadal WAT (gWAT) (Fig. 1b).

### Mast cell serotonin inhibits WAT browning in vitro. To identify cell types that may contribute to the increases in WAT Tph1, we first queried the BIOGPS gene atlas[21] for genes that were associated with Tph1 expression and found 12 highly correlated genes (>0.95 $R^2$ value). Notably, seven of these genes were known to be highly enriched in mast cells (Kit, FCER1a, Cpa3, Tpsg1, Cma1, Alox5, and Ms42a) (Fig. 1c, Supplementary Table 1). However, this query did not identify genes associated with adipocytes as previously indicated[14], suggesting that mast cells may be the predominant cell type contributing to serotonin production in adipose tissue.

Mast cells are immune cells that reside within tissues to quickly respond to injury or other pro-inflammatory stimuli, secreting cytokines, proteases and bioamines to potentiate an inflammatory response[22]. Consistent with increases in WAT Tph1 at thermoneutrality, we found increased expression of the mast cell marker tryptase β2 (Tpsb2), which was correlated with increased Tph1 expression (Fig. 1d, e). Increased expression of both Tpsb2 and Tph1 at thermoneutrality was associated with elevated serotonin levels in both WAT depots (Fig. 1f), an effect independent of changes in whole blood serotonin (Supplementary Fig. 1g). These data indicate that thermoneutrality increases mast cells within WAT and this is associated with increases in Tph1 and serotonin.

To examine whether there might be a causal link between mast cells, Tph1, serotonin and reduced WAT Ucp1 at thermoneutrality, we cultured mast cells in vitro and treated them with the Tph chemical inhibitor LP533401[23] followed by the calcium ionophore, A23187, to induce mast cell degranulation (Fig. 1g). As expected, A23187 treatment increased serotonin release (Fig. 1h), however, LP533401 pre-treatment dramatically reduced this effect (Fig. 1h). To examine whether this serotonin production from mast cells could directly inhibit Ucp1 expression in WAT, we subsequently cultured iWAT stromal vascular cells and treated them with 1μM of serotonin starting at the beginning of differentiation (day 7) (Fig. 1i). Treatment of these cells with the pan-β-adrenergic agonist isoproterenol increased Ucp1, but this was mitigated by serotonin treatment (Fig. 1j). Collectively, these data suggest that at thermoneutrality, there are increases in the number of mast cells, Tph1 and serotonin which can inhibit isoproterenol-induced increases in Ucp1 in WAT.

### Mast cell Tph1 promotes obesity & insulin resistance. Previous studies have found that mast cells accumulate within obese WAT of both mice[24] and humans[25]. To examine whether mast cell serotonin contributes to obesity and insulin resistance, mice lacking functional mast cells (Kit[W-sh/W-sh]) were injected with saline (Kit[sham]) or in vitro-cultured bone marrow-derived mast cells (BMMCs) from Tph1[+/+] (Kit[Tph1+/+]) or Tph1[−/−] (Kit[Tph−/−]) mice and fed a HFD (Fig. 2a). Flow cytometry analysis (Supplementary Fig. 2a) using established markers of mast cell maturity (CD117[+]/FcεR1[+]) indicated that there were no differences in BMMC viability or purity between genotypes (Supplementary Fig. 2b) and, as expected, Tph1 expression was dramatically reduced (~99.9%) in mast cells of Tph1[−/−] mice (Fig. 2b). Additionally, Tph2 was nearly undetectable in both Tph1[+/+] or Tph1[−/−] mast cells (Fig. 2b) and the expression of

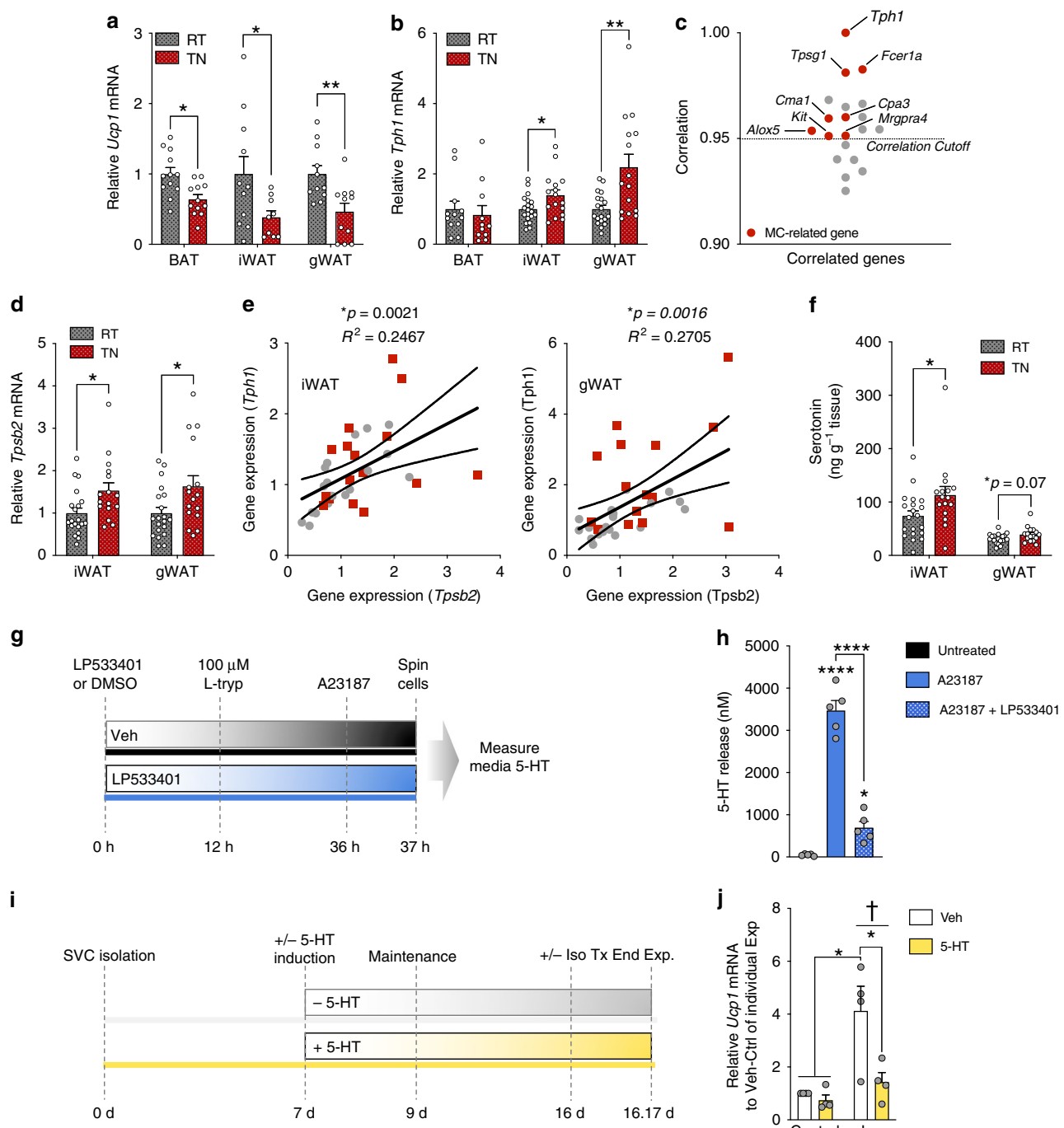

**Fig. 1 Thermoneutrality reduces white adipose tissue *Ucp1* and increases *Tph1*, mast cells and serotonin in HFD-fed mice. a** *Ucp1* expression in BAT ($n = 12$), iWAT ($n = 11$ RT, 9 TN) and gWAT ($n = 11$). **b** *Tph1* expression in BAT ($n = 12$ RT, 11 TN), iWAT ($n = 20$ RT, 16 TN) and gWAT ($n = 19$ RT, 15 TN). **c** BIOGPS *Tph1* Correlations highlighting mast cell-related genes with greater than 0.95 cutoff. **d** *Tpsb2* expression in WAT ($n = 20$ RT, 16 TN). **e** Correlation between *Tpsb2* and *Tph1* expression of RT (grey dots) and TN (red squares) housed mice in iWAT ($n = 36$) and gWAT ($n = 34$) Slopes were determined to be significantly (\*$p < 0.05$) non-zero by simple linear regression. **f** Serotonin (5-HT) in WAT ($n = 20$ RT, 16 TN) **g** Timeline of in vitro experiment. L-tryp, L-tryptophan. **h** Serotonin release stimulated with A23187 (2 μM) in vitro-cultured mast cells with or without 300 μM LP533401 ($n = 5$) as determined by One-way ANOVA and Tukey post hoc test. Significance (\*$p < 0.05$) compared to Untreated group, unless indicated otherwise. **i** Beige adipocyte progenitors were treated with vehicle (Veh) or 1 μM serotonin (5-HT) starting at the beginning of differentiation and terminated after a 4-hour exposure to 10 nM isoproterenol (Iso). **j** *Ucp1* expression in primary cultured beige adipocytes ($n = 4$). Significant effects (\*$p < 0.05$) and isoproterenol treatment effects ($^{\dagger}p < 0.05$) determined by two-way ANOVA with uncorrected Fisher's LSD post-test. RT-TN statistical significance assessed via Student's *t*-test. Significance denoted by \*$p < 0.05$, \*\*$p < 0.01$, \*\*\*\*$p < 0.0001$). Data are expressed as mean ± SEM.

the serotonin transporter did not differ between Tph1$^{+/+}$ or Tph1$^{-/-}$ mast cells (Supplementary Fig. 2c).

As others have shown[26,27], Kit$^{sham}$ mice fed a HFD had attenuated weight gain and adiposity relative to controls, a

phenotype that was reversed following the injection of mast cells from Tph1$^{+/+}$ (Kit$^{Tph1+/+}$) donors (Fig. 2c, d). However, this increase in weight gain and adiposity was not observed in mice injected with mast cells from Tph1$^{-/-}$ donors (Fig. 2c, d,

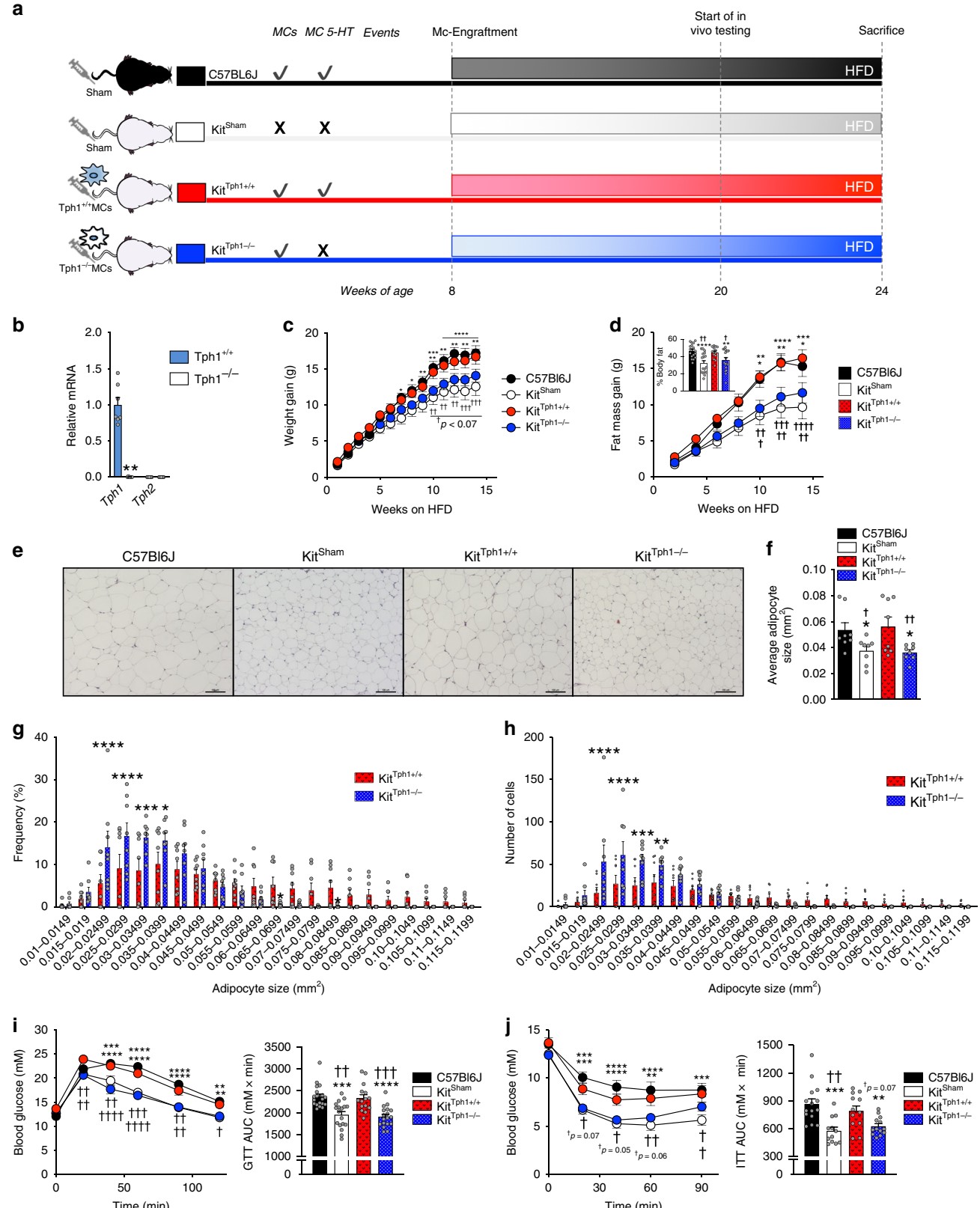

Supplementary Fig. 2d). Consistent with reduced adiposity, iWAT from Kit$^{Tph1-/-}$ mice had smaller adipocytes (Fig. 2e, f), that occurred with a higher frequency (Fig. 2g) and number (Fig. 2h) compared to Kit$^{Tph1+/+}$ mice, a finding similar to C57BL6J and Kit$^{sham}$ controls (Supplementary Fig. 2e, f). Consistent with their reduced adiposity, Kit$^{Tph1-/-}$ mice had improved glucose tolerance (Fig. 2i) and insulin sensitivity (Fig. 2j) compared to controls. Importantly, these differences in metabolism between Kit$^{Tph1+/+}$ and Kit$^{Tph1-/-}$ mice were not due to differences in the number of mast cells within iWAT and gWAT, which were comparable between genotypes (Supplementary Fig. 2g, h).

**Fig. 2 Mast cells are enriched in Tph1 and HFD-fed mice lacking mast cell serotonin are protected from obesity and insulin resistance. a** HFD-fed C57BL6J ($n = 19$) and mast cell-deficient mice were engrafted with either sham, Tph1$^{+/+}$ or Tph1$^{-/-}$ mast cells beginning at 8 weeks of age for 12 weeks and in vivo tests following. The sample size indicated is for all figures unless otherwise stated. Significant effects of C57BL6J mice (*$p < 0.05$) and Kit$^{Tph1+/+}$ (†$p < 0.05$) between Kit$^{Sham}$ and Kit$^{Tph1-/-}$ as determined by two-way RM ANOVA or one-way ANOVA with uncorrected Fisher's LSD post-test. **b** Relative expression of Tph1 and Tph2 from in vitro-cultured mast cells from Tph1$^{-/-}$ mice or wildtype littermates ($n = 5$) determined by Student's $t$-test. **c** Weight gain in grams over 14 weeks of HFD of C57BL6J ($n = 19$), Kit$^{Sham}$ ($n = 17$), Kit$^{Tph+/+}$ ($n = 15$) and Kit$^{Tph-/-}$ ($n = 18$). **d** Fat mass gain in grams over 14 weeks and % Body fat at 12 weeks of HFD of C57BL6J ($n = 19$), Kit$^{Sham}$ ($n = 17$), Kit$^{Tph+/+}$ ($n = 15$) and Kit$^{Tph-/-}$ ($n = 18$). **e** Representative H & E images of mast cell-engrafted Kit$^{W-sh/W-sh}$ mice iWAT, scale bars set at 100 μm. **f** Average adipocyte sizes of mast cell-engrafted Kit$^{W-sh/W-sh}$ mice iWAT ($n = 8$). **g** Frequency of adipocytes in iWAT of Kit$^{Tph+/+}$ and Kit$^{Tph-/-}$ mice ($n = 6$). Significance (*$p < 0.05$) determined by two-way ANOVA with uncorrected Fisher's LSD post-test. **h** Number of adipocytes in iWAT of Kit$^{Tph+/+}$ and Kit$^{Tph-/-}$ mice ($n = 6$). Significance (*$p < 0.05$) determined by two-way ANOVA with uncorrected Fisher's LSD post-test. **i** Glucose tolerance test (GTT), 1 g/kg glucose. **j** Insulin tolerance test (ITT), 0.7 U/kg insulin. ($n = 15$ C57BL6J, 12 Kit$^{Sham}$, 12 Kit$^{Tph+/+}$, and 13 Kit$^{Tph-/-}$). Significance denoted by *$p < 0.05$, **$p < 0.01$, ***$p < 0.001$, ****$p < 0.0001$. All data are expressed as mean ± SEM.

To further substantiate the observations that mast cell Tph1 promotes obesity and insulin resistance without the potential confounding metabolic effects of the Kit$^{W-sh/W-sh}$ mutation[28], we removed Tph1 in mast cells by crossing Tph1 floxed mice with mice expressing Cre recombinase linked to carboxypeptidase 3 promoter (Cpa3-Cre) to generate Tph1 mast cell null mice (Tph1 MCKO) (Fig. 3a) and confirmed deletion in intraperitoneal mast cells (Supplementary Fig. 2i). Tph1 MCKO mice were then fed a HFD starting at 8 weeks of age and similar to observations with Kit$^{Tph1-/-}$ mice, we found that Tph1 MCKO mice were also protected from HFD-induced weight gain (Fig. 3b, Supplementary Fig. 2j) due to reductions in adiposity (Fig. 3c, d). Tph1 MCKO mice were more glucose tolerant (Fig. 3e), insulin sensitive (Fig. 3f) and displayed a reduction in fasting blood glucose (Fig. 3g), liver weight (Fig. 3h) and liver lipids (Fig. 3i, j). These changes in metabolism were independent of alterations in free serotonin levels measured from platelet-poor plasma (Fig. 3k). These differences between genotypes were also independent of changes in markers of mast cells (Tpsb2) or other immune cell types such as basophils (Basoph8), M1 macrophages (Nos), eosinophils (SiglecF), M2 macrophages (Arg1) and ILC2s (Gata3) (Supplementary Fig. 2k). These data indicate that mice lacking Tph1 in mast cells are protected from obesity and insulin resistance independently of alterations in circulating serotonin.

**Mast cell serotonin suppresses energy expenditure & WAT Ucp1.** To examine the potential mechanisms contributing to reduced weight gain in the absence of mast cell Tph1, we assessed caloric intake and energy expenditure using indirect calorimetry. In the Kit model, food (Supplementary Fig. 3a) and water intake (Supplementary Fig. 3b) were comparable between genotypes. However, resting VO$_2$ (oxygen consumption during times of <30 beam breaks per 18-minute interval) (Fig. 4a) and 24 h VO$_2$ (Supplementary Fig. 3c, d) was elevated in both Kit$^{sham}$ and Kit$^{Tph1-/-}$ mice compared to controls. This increase in VO$_2$ was present independent of differences in body mass (Supplementary Fig. 3e) and was not due to differences in physical activity levels, which were comparable between genotypes (Supplementary Fig. 3f). Similar to our studies in Kit$^{Tph1-/-}$ mice, Tph1 MCKO mice had no change in physical activity, food or water intake (Supplementary Fig. 3g–i), but did have increases in resting (Fig. 4b) and 24 h VO$_2$ (Supplementary Fig. 3j, k), however, this was measured at a time when there were differences in body mass (Supplementary Fig. 3l). Collectively, these data demonstrate that the deletion of Tph1 in mast cells increases resting energy expenditure.

To discern the tissues involved in enhanced energy expenditure in mice lacking mast cell Tph1, we first conducted assays focused on the potential role of BAT. Using an infrared thermography assay which directly detects Ucp1-mediated thermogenesis in

BAT[29], we found no differences in interscapular surface temperatures between Kit$^{Tph1+/+}$ and Kit$^{Tph1-/-}$ mice following the injection of saline or CL-316,243 (Supplementary Fig. 4a). Consistent with this observation, Ucp1 mRNA (Supplementary Fig. 4b), Ucp1 protein expression (Supplementary Fig. 4c), BAT mass (Supplementary Fig. 4d) and morphology (Supplementary Fig. 4e) were comparable between Kit$^{Tph1+/+}$ and Kit$^{Tph1-/-}$ mice. Similarly, Tph1 MCKO mice also had no discernable difference from controls with respect to BAT mass and morphology (Supplementary Fig. 4f–h). These data indicate that mast cell serotonin is unlikely to elicit its beneficial effects on body mass and adiposity through modulation of BAT thermogenesis; a finding consistent with the observation that thermoneutrality does not alter Tph1 expression in BAT (Fig. 1b).

In contrast to BAT, the iWAT of Kit$^{Tph1-/-}$ mice had reductions in Tph1 expression and adipose tissue serotonin (Fig. 4c, d) and increased expression of Ucp1 (Fig. 4e) compared to Kit$^{Tph1+/+}$ mice. Consistent with these observations, iWAT and gWAT from Tph1 MCKO mice tended and had significantly lower Tph1 mRNA, respectively (Fig. 4f). These reductions in Tph1 expression were associated with greater multilocular lipid droplet formation (Fig. 4g, top panel), greater Ucp1 protein (Fig. 4g, bottom panel) and mRNA (Fig. 4h) in iWAT and in gWAT (Fig. 4i, j). Similar findings with respect to the browning of iWAT has also recently been reported in mice fed a control chow diet[30].

## Discussion

Mast cells accumulate in target tissues in response to allergic or inflammatory stimuli and secrete various signaling molecules such as cytokines, proteases and bioamines[22]. Previous studies have shown that mast cells accumulate within the WAT of both obese mice[24] and humans[25]. We now show that when mice are housed at thermoneutrality, a condition known to reduce Ucp1 and adipose tissue thermogenesis[20], there is an increase in mast cells within WAT that occurs independent of changes in adiposity. This increase in mast cells is associated with increased WAT Tph1 expression and serotonin. Using two distinct mouse models we subsequently demonstrate that genetic deletion of Tph1 in mast cells reduces WAT serotonin and enhances WAT Ucp1 and whole-body energy expenditure; an effect associated with a reduction in high-fat diet-induced obesity, insulin resistance and NAFLD. These data establish a critical role for mast cell serotonin synthesis in suppressing adipose tissue thermogenesis and suggest that therapeutic targeting of this pathway may exert beneficial metabolic effects in obesity.

In the current study, the increase in energy expenditure that we[13] and others[14] have observed with whole-body inhibition of Tph1 and subsequent reductions in circulating serotonin can be recapitulated by removing Tph1 in mast cells. These findings

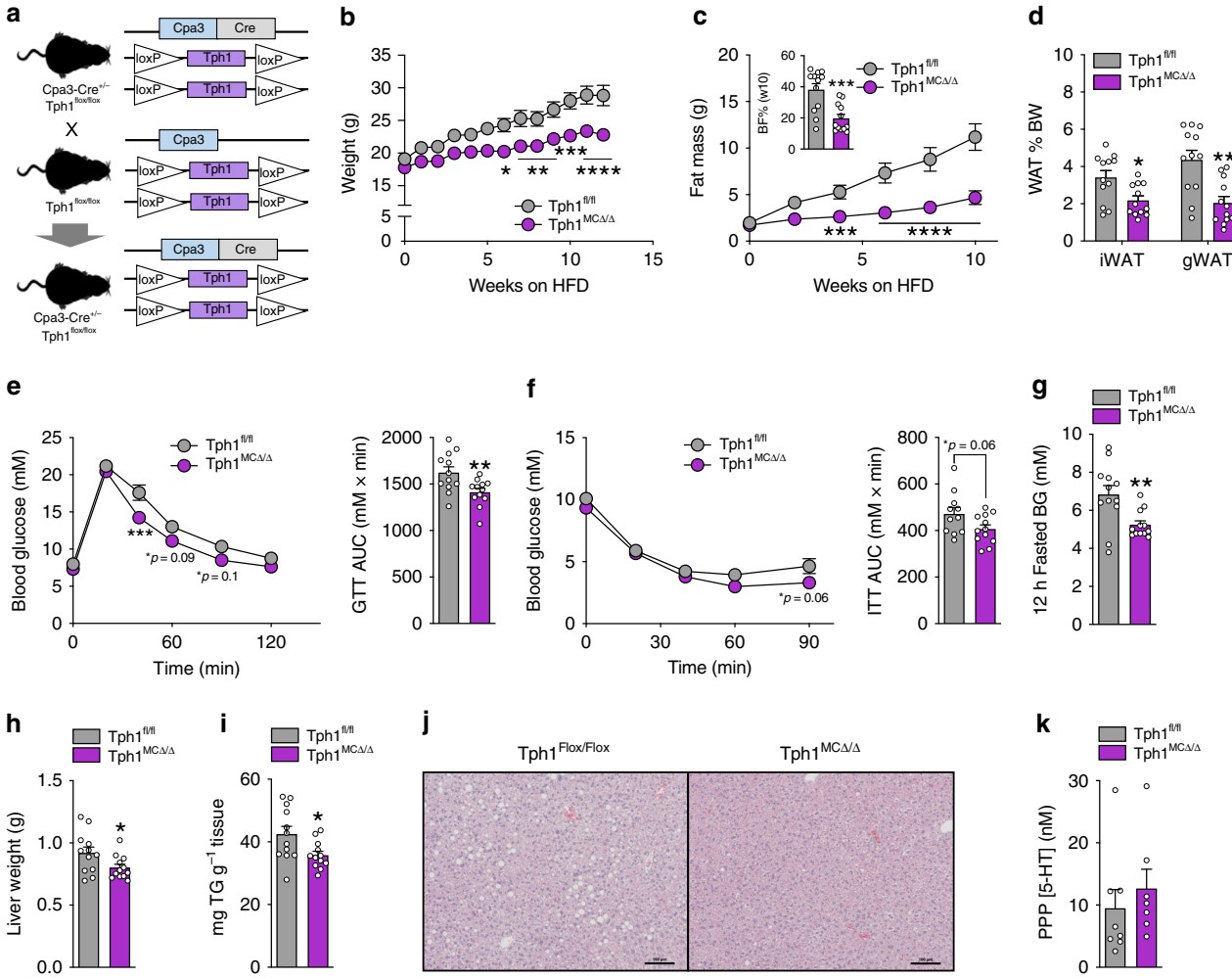

**Fig. 3 Mast cell-specific Tph1 deletion protects HFD-fed mice from obesity and insulin resistance. a** Breeding strategy to produce Carboxypeptidase-3 (Cpa3) Cre Tph1 double floxed mice. **b** Weight and **c** Fat mass over time ($n = 12$). **d** WAT mass % of body weight ($n = 12$). **e** Glucose tolerance test (GTT), 1.5 g/kg glucose ($n = 12$). **f** Insulin tolerance test (ITT), 1 U/kg insulin ($n = 12$). **g** Overnight 12 h fasted blood glucose (BG, $n = 12$). **h** Liver weight in grams ($n = 12$). **i** mg of Triglycerides (TG) per gram of liver tissue ($n = 12$). **j** Representative liver H & E images, scale bar 100 μm. **k** Serotonin concentration in platelet poor plasma (PPP) samples ($n = 8$ Tph1[fl/fl], 7 Tph1 MCKO). Statistical significance (*$p < 0.05$, **$p < 0.01$, ***$p < 0.001$, ****$p < 0.0001$) determined by two-way RM ANOVA with Bonferroni post-test or Student's $t$-test. All data are expressed as mean ± SEM.

seem paradoxical based on previous findings indicating that serotonin in the central nervous system increases BAT thermogenesis[31–34]. These opposing functions of central and peripheral serotonin are consistent with findings from other highly conserved regulators of energy balance such as the AMP-activated protein kinase where genetic reductions of hypothalamic AMPK increases energy expenditure[35], while reductions of AMPK in adipose tissue lower energy expenditure and iWAT browning[36]. A similar paradigm of opposing functions between central and adipose specific mTOR has also been observed[37–39]. Thus, there is a precedent by which central and peripheral pathways may exert opposing functions on energy expenditure. Collectively, these data support a model where Tph2 and central serotonin enhance energy expenditure in response to cold, but under thermoneutral conditions or with obesity, peripheral serotonin synthesis by mast cell Tph1 reduces thermogenic activation, thus lowering adipose tissue energy expenditure.

However, a fundamental question is why would mast cell serotonin production by Tph1 have evolved to suppress adipose tissue thermogenesis? Evolution inference has been shown to inform biological function[40]. Querying the Clustering by Inferred

Models of Evolution (CLIME) portal (http://gene-clime.org/) for genes that are highly likely to have coevolved with Tph1 reveals, remarkably, only two hits: (1) the highly related orthologue Tph2 and (2) tyrosine hydroxylase, the primary enzyme regulating norepinephrine production and adipose tissue thermogenesis. As mast cells are the first responders to injury and primary cell type initiating anaphylaxis and inflammation[22], it is interesting to speculate that the suppression of adipose tissue thermogenesis by mast cell serotonin may have developed to reduce futile cycling so that substrate could be redirected towards other immune functions. Future studies investigating how mast cell serotonin reduces Ucp1, including the primary serotonin receptor and downstream signaling, and whether mast cell serotonin inhibits Ucp1-independent thermogenesis[41,42], will be important for further understanding the mechanisms linking mast cell Tph1 and adipose tissue thermogenesis.

Our findings also raise important questions about the role of mast cells in obesity and insulin resistance and the potential limitation of mouse models of mast cell deficiency to establish causality. Recent studies by Gutierrez and colleagues[28] indicated that previous findings linking mast cells with obesity-induced

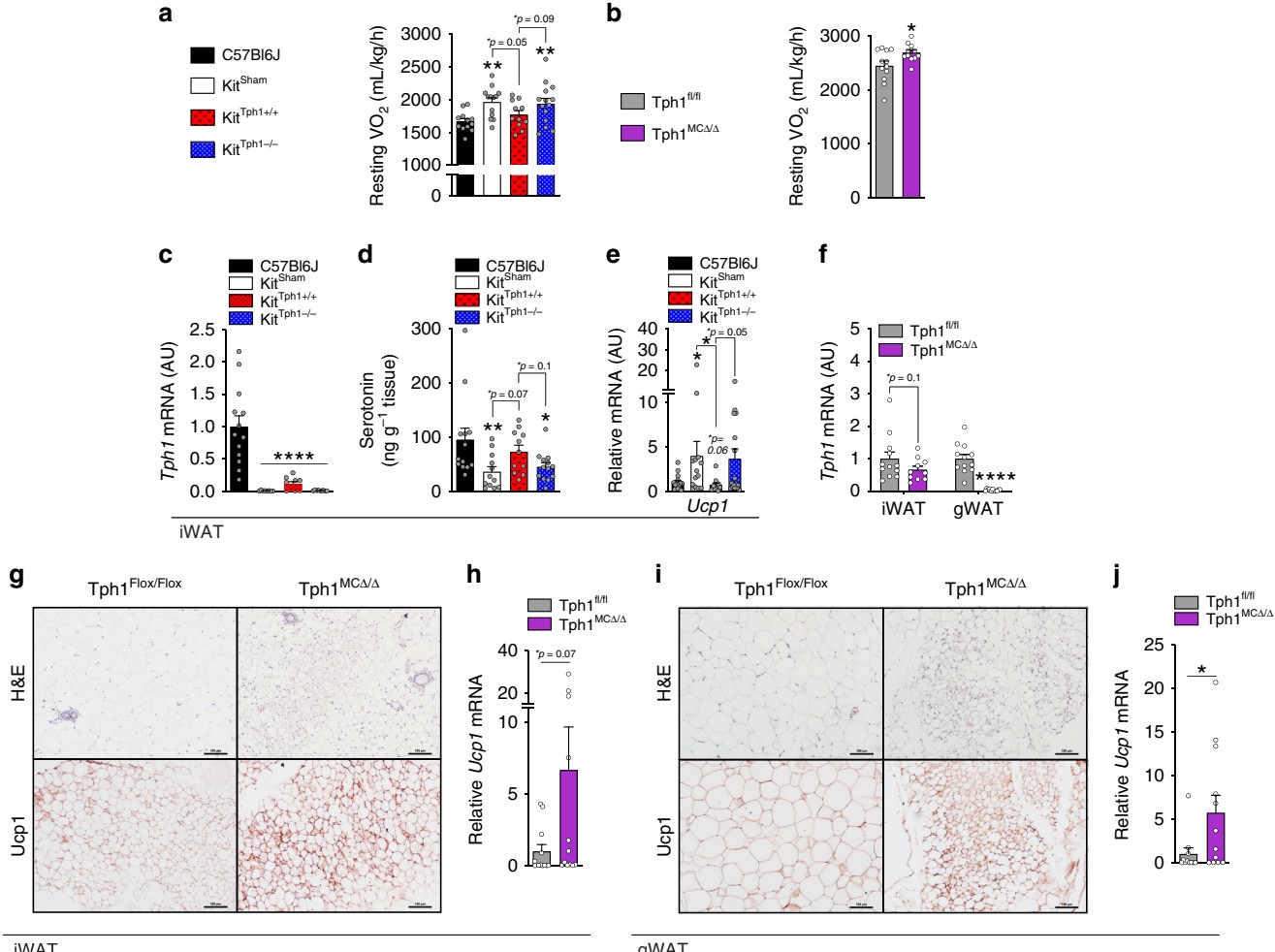

**Fig. 4 Genetic removal of mast cell serotonin increases energy expenditure and causes adipose tissue browning. a** Resting oxygen consumption of C57BL6J ($n = 12$), Kit$^{Sham}$ ($n = 13$), Kit$^{Tph+/+}$ ($n = 11$) and Kit$^{Tph-/-}$ ($n = 14$). **b** Resting oxygen consumption of Tph1 MCKO mice ($n = 12$). **c** Tph1 expression in iWAT of C57BL6J ($n = 13$), Kit$^{Sham}$ ($n = 8$), Kit$^{Tph+/+}$ ($n = 9$) and Kit$^{Tph-/-}$ ($n = 8$). **d** Adipose tissue serotonin (5HT) of C57BL6J ($n = 13$), Kit$^{Sham}$ ($n = 12$), Kit$^{Tph+/+}$ ($n = 11$) and Kit$^{Tph-/-}$ ($n = 13$). **e** Ucp1 expression of C57BL6J ($n = 15$), Kit$^{Sham}$ ($n = 14$), Kit$^{Tph+/+}$ ($n = 12$) and Kit$^{Tph-/-}$ ($n = 16$). **f** Tph1 expression in WAT of Tph1 MCKO mice ($n = 12$). **g** Representative images of H & E and Ucp1 IHC-stained iWAT of Tph1 MCKO mice, Scale bar 100 μm. **h** Ucp1 expression in iWAT from Tph1 MCKO mice ($n = 12$). **i** Representative images of H & E and Ucp1 IHC-stained gWAT of Tph1 MCKO model, Scale bar 100 μm. **j** Ucp1 expression of gWAT from Tph1 MCKO model ($n = 12$). Statistical significant effects (*$p < 0.05$, **$p < 0.01$ and ****$p < 0.0001$) determined by one-way ANOVA with uncorrected Fisher's LSD post-test or Student's $t$-test where appropriate. All data are expressed as mean ± SEM.

insulin resistance[27] may have been due to effects intrinsic to the Kit$^{W-sh/W-sh}$ mutation rather than a direct result of mast cell reduction. These conclusions were based on the findings that: (1) the restoration of hematopoietic cells in Kit$^{W/Wv}$ mice, whether or not the hematopoietic cells could generate mast cells, led to weight gain when mice were fed a HFD and (2) mice containing an overexpressing Cre recombinase linked to Cpa3, which results in the death of mast cells, are not protected from HFD-induced obesity or insulin resistance. It should be noted that our studies differ in many ways from those of Gutierrez[28] or Liu and colleagues[27] as we did not directly test the role of mast cells per se in the regulation of obesity, but rather made comparisons between mast cells with or without Tph1 using two unique and completely distinct mouse models. First, we showed that reconstitution of Kit$^{W-sh/W-sh}$ mice with Tph1$^{+/+}$ mast cells increased weight gain and insulin resistance, while Tph1$^{-/-}$ mast cells did not. It is important to note this difference in weight gain between Kit$^{Tph1+/+}$ and Kit$^{Tph1-/-}$ mice was not due to differences in the

number of mast cells within WAT, which were comparable upon reconstitution. Secondly and perhaps most importantly, given the potential confounding effects of the Kit$^{W-sh/W-sh}$ mutation on metabolism[28], we generated an entirely distinct mouse model in which Tph1 floxed mice were crossed to Cpa3$^{Cre/+}$ mice and observed similar results with respect to weight gain, adipose tissue Ucp1, energy expenditure and insulin sensitivity in mice lacking mast cell Tph1. And while it is known that Cpa3 is also expressed in basophils[43], it is important to note that mast cells have a much higher expression of Tph1 (>1000-fold) compared to basophils or other immune cells types such as macrophages making it unlikely that these cell types contributed to the phenotype[44]. Therefore, our findings, in two distinct mouse models as suggested[28,45] support the conclusions that mast cell Tph1 protects against obesity induced insulin resistance by enhancing adipose tissue Ucp1 and whole body energy expenditure.

While these models are complementary, there are important differences in our results. The first and most evident is that the

protection from obesity-induced glucose intolerance and insulin resistance is more dramatic in the Kit[Tph1−/−] mice (Fig. 2) compared to the Tph1 MCKO mice (Fig. 3). This difference is most likely attributed to sex differences between models; since males were studied in the Kit[W-sh/W-sh] mast cell reconstitution experiments while female mice were studied in the Tph1 MCKO experiments. As female mice are lighter and more insulin sensitive than male mice at the same age and duration of high-fat diet feeding[46], this likely underlies the primary difference between the models. Other reasons for the more modest phenotype in the Tph1 MCKO mice is the time of testing which was after 8 weeks of high-fat diet compared to the 12 weeks for the Kit[Tph1−/−] mice. The longer duration of high-fat diet feeding in the Kit[Tph1−/−] mice was utilized to maximize mast cell engraftment as previously suggested[47]. Therefore, sex and duration of the high-fat diet before testing may explain the differences between models. However, we believe showing a similar phenotype in two different sexes, albeit to a lesser degree in the female mice, is an important strength of the study.

Another important consideration when interpreting our findings on energy expenditure are that serotonin is a potent regulator of vascular tone (reviewed here ref. [48]). The regulation of vasculature tone is extremely complex and depending on the tissues and serotonin receptors expressed, serotonin can induce either vasodilation or vasoconstriction to regulate heat loss and heat retention, respectively (reviewed here ref. [48]). Mice with germline deletions of Tph1 have altered blood pressure and vascular tone[49] and chemical inhibition of Tph1 can reduce hypoxia-induced pulmonary-arterial hypertension[50]. However, we have shown that in obese mice fed a HFD, similar to the current study, that treatment with the pan-Tph inhibitor, LP533401, does not affect heart rate or blood pressure, and that the effects of this compound on weight loss and insulin sensitivity are completely dependent on the expression of Ucp1[13]. These data suggest that in the context of high-fat diet-induced obesity it is unlikely that genetic deletion of Tph1 specifically in mast cells, which importantly had no effect on circulating serotonin, would elicit any effect on vasculature tone or heat loss. Lastly, while human mast cells synthesize serotonin via Tph1[51], it is important to highlight that there are relatively small amounts of serotonin secreted and stored in human mast cells in comparison to rodents. Future studies investigating the role of mast cell serotonin synthesis in humans and the potential impact this may have on vasculature tone are warranted.

In conclusion, our findings identify a role for mast cell serotonin for energy balance and suggest that inhibition of mast cell Tph1 may represent a promising strategy for the targeted treatment of obesity and insulin resistance. Future studies examining the importance of this pathway in humans with obesity and insulin resistance are warranted.

## Methods

**Animal housing and breeding.** All animal experiments were performed in accordance with the McMaster Animal Care Committee guidelines and conducted under the Canadian guidelines for animal research (AUP: 16-12-41) and all study protocols received ethical approval. Tph1KO and WT littermates on C57BL6J background have been characterized previously that are bred in house[13,52]. Tph1 mast cell-specific knockout (Tph1 MCKO) mice were generated by crossing Tph1 floxed mice[53] with mice expressing Cre-Recombinase under the control of Carboxypeptidase-3 (Cpa3-cre)[43]. Kit[W-sh/W-sh] mice and Cpa3-cre (Stock No: 012861 & 026828, respectively) mice on a C57BL6J background were purchased from Jackson Laboratories and have been characterized previously[43,54].

**Cell in vitro experiments.** Male 6–16 week old Tph1[+/+] and Tph1[−/−] littermates were sacrificed and bone marrow from the tibia and femur were used to culture bone marrow-derived[47]. Briefly, mice were sacrificed by cervical dislocation and bone marrow was extracted from the femur and tibia by spinning the bones with severed ends. The resulting bone marrow was strained through a 10 μm strainer

and topped up to 10 mL with DMEM media containing 10% FBS, 1% L-glutamine, 1% antibiotic (penicillin/streptomycin), 10 ng/mL recombinant mouse IL-3 (Peprotech, 213-13) and 50 μM β-mercaptoethanol. Media was changed every 3–4 days for 8–10 weeks with regular maturation checks with Kimura dye[55] within the first 6 weeks of culture. Maturation was confirmed by FACS using c-Kit (CD117) and FCeRI antibodies. For cell experiments, bone marrow-derived mast cells were counted and plated into 24-well plates (250,000 cells). For dose response and 5-HT release experiments, 1–1000 μM of LP533401 or DMSO vehicle were administered for 12 h and subsequently treated with 100 μM L-tryptophan for 24 h. A calcium ionophore that degranulates bone marrow-derived mast cells (A23187) was administered (2 μM) for 1 hour before cells were spun down (200 g) and media collected. For intraperitoneal lavage of mast cells, mice were sacrificed by cervical dislocation and 10 mL of PBS was injected into the intraperitoneal space of the mouse. The abdomen of the mouse was massaged and the contents drawn into a syringe, which was spun down at 2400 g for 10 min and used for RT-qPCR.

**Animal experiments.** Mice were maintained on a 12-h light and dark cycle (lights on at 0700) with food and water provided *ad libitum*. For experiments conducted in Fig. 1, male mice were housed at either room temperature (23 °C) or thermoneutrality (29 °C) and were fed a 60% HFD starting at 8 weeks of age for 10 weeks. All other experiments were performed in mice housed solely at room temperature. For bone marrow-derived mast cell reconstitution experiments (related to Figs. 2 and 4) male Kit[W-sh/W-sh] mice[47] at 9–10 weeks of age were separated into three different weight-matched groups and were injected with a total of 20 million mast cells from Tph1[+/+] (Kit[Tph1+/+]) or Tph1[−/−] (Kit[Tph1−/−]) mice at three different anatomical locations (10 million intravenously through tail vein, 5 million subcutaneously (dorsal interscapular area) and 5 million intraperitoneally). The third group of Kit[W-sh/W-sh] mice received an injection of equal volume saline at all three sites (Kit[sham]). C57BL6J mice were also injected with equivalent volume of saline at all aforementioned sites. After injections, mice were fed a 45% HFD (Cedarlane Canada; D12451) for 16 weeks (kept in animal facility for 24–25 weeks) with tissues collected after a 6-h fast and administration of 0.7 U/kg intraperitoneal insulin, 15 min prior to sacrifice by cervical dislocation. Female Tph1 MCKO mice (related to Figs. 3 and 4) were fed a HFD at 8 weeks of age for 12 weeks (kept in animal facility for 20 weeks) and sacrificed after a 6-h fast. Platelet poor plasma was extracted by pipetting 152 μL of whole blood into 8μL of 0.5 M EDTA, spun at 1500 g and top 20 μL of plasma removed. Liver, iWAT, gWAT and BAT from all mice were weighed and samples were kept at −80 °C.

**Metabolic testing.** Mice were weighed weekly and body composition analyzed biweekly using a Bruker minispec Whole Body Composition Analyzer. For Kit experiments mice were subjected to a 1 g/kg GTT and 0.7 U/kg ITT at 12 and 14 weeks post mast cell engraftment. For Tph1 MCKO experiments mice were subjected to 1.5 g/kg GTT and 0.75 U/kg ITT after 8 and 9 weeks of HFD. For both ITT and GTT experiments animals were fasted for 6 h starting at 0700 hours and basal blood glucose concentrations were measured using a glucometer (Aviva, Roche) at indicated time points. Measurements of metabolic parameters (respiratory exchange ratio, energy expenditure, activity, food and water intake) were assessed using a Comprehensive Laboratory Animal Monitoring System (CLAMS; Columbus Instruments) over a 48-h period.

**Infrared imaging of Ucp1-mediated thermogenesis.** Ucp1-mediated thermogenesis was assessed using an infrared imaging technique[29]. Briefly, mice are anesthetized with Avertin made with 25 g of 2,2,2-tribromoethanol (Sigma-Aldrich, #4, 840-2) in 15.5 mL tert-amyl alcohol to form a 80x stock solution and diluted to 20 mg/mL in saline. After 2 min post-Avertin injection, mice are injected intraperitoneally with either saline or CL-316,243 (0.0155 mg/kg, R&D Systems, Inc; Item # 1499/10). Average of oxygen consumption taken at 5 s intervals with an indirect calorimetry system (CLAMS, Columbus Instruments, Columbus, OH) from 19–20 min post Avertin injection and the top 10% interscapular dorsal temperature is determined by a thermal image taken 20 min post-Avertin by infrared camera (T650sc, emissivity of 0.98, FLiR Systems). Thermal images are analyzed using AMIDE-bin 1.0.5 software. Experiment is conducted on two different days (separated by 2–3 days between tests) on the same mice for saline and CL-316,243 injections.

**Immunoblotting analysis.** Tissues were placed in lysis buffer (20 mM Tris, 150 mM NaCl, 1 mM EDTA, 1 mM EGTA, 1% Triton-X, 2.5 mM sodium pyrophosphate, 0.5 mM DTT, 0.1% SDS, 1% Roche Protease Inhibitor, 0.5% sodium deoxycholate) and subsequently processed in a tissue homogenizer (Percellys). Samples were spun to extract supernatant and protein quantification was performed using the bicinchonininc acid (BCA) method (Pierce) as per manufacturer instructions. For UCP1, β-tubulin, β-actin protein analysis, BAT or iWAT lysates were diluted with 4× standard buffer (50% Sucrose, 7.5% SDS, 3.1% DTT) and loaded in Sodium dodecyl sulfate-polyacrylamide gel electrophoresis (SDS–PAGE) was performed using 10 or 12% gels. A Western Blotting apparatus (Bio-Rad) with electrophoresis buffer containing 12.5 mM Tris, 125 mM Glycine, 0.05% SDS. About 20 μg of sample was dispensed into wells along with Precision Plus Protein Dual Colour Standard (Bio-Rad). A wet transfer using a gel to membrane blotter

apparatus (BioRad) of PVDF membranes, 90 V for 90 min. Membranes were blocked with 5% skim milk in Tris-buffered saline (50 mM Tris–HCl, 150 mM NaCl) with Tween 20 (TBST) for 1 h. Subsequently, membranes were subject to primary antibody (1:1000 dilution) in TBST + 5% BSA overnight at 4 °C. Antibodies used were Anti-Mouse UCP1 (Alpha Diagnostics, 173435A4) and Anti-Mouse β-tubulin (Invitrogen, 322600). Membranes were rinsed in TBST and incubated in secondary antibody (1:10000 dilution in TBST + 5% BSA; Anti-mouse, 7076 S; Anti-rabbit, 7074 S) for 1 h. SuperSignal West Femto Maximum Sensitivity Substrate (Thermo Scientific) and a Fusion FX7 Chemiluminescence Visualizer (MBI) were used for detection. Densitometry analysis was performed using an ImageJ 3 analyzer.

**Histological analysis.** Tissues were immersed in 10% formalin, 90% PBS for 1 day and placed into 70% EtOH for paraffin embedding and H & E staining by the John Mayberry Histology facility at McMaster University. Adipocyte size assessment of H & E and immunohistochemistry for Ucp1 staining were done[56]. Briefly, H&E slides were taken at 10x magnification. A measurement of the farthest diameter for each adipocyte in the picture was used for data analysis using ImageJ. Images that filled the entire picture were used for this analysis. For Ucp1 staining, paraffin embedded slides are deparaffinized in xylenes. After 100% EtOH wash, endogenous peroxidase activity was blocked with hydrogen peroxide solution in 100% MeOH, followed by a 70% EtOH and dH₂O washes. Slides are placed in a pressure cooker at high temperature for 5 min in citrate buffer, cooled and washed in Tris buffer. Slides are blocked with 5% goat serum in Tris buffer. Primary antibody (Ucp1, 173435A4, Alpha Diagnostics) concentration at 1:200 was used followed by 1:500 of secondary antibody. Stepdavidin peroxidase (1:50, MJS BioLynx Inc.) is used followed by Nova red (MJS BioLynx Inc.) as per manufacturer's protocol. Images were captured with a light 90 Eclipse microscope (Nikon). For CD117 staining, tissues were sectioned at 4 microns on a microtome (Leica). After deparaffinization and re-hydration of the sections, we performed epitope retrieval using a pressure cooker (Cuisinart) and citrate buffer. We then used a mouse on mouse staining kit with an HRP-based Nova Red developing reagent (both Vector Labs). Blocking was performed with normal goat serum and secondary only control slides were included to confirm staining specificity. Slides were then cover slipped using Permount and later imaged using a Lifetech EVOS XL microscope.

**RNA extraction.** Tissue samples (50–100 mg) are homogenized in 1 mL of Trizol reagent (Life Technologies). Samples are spun for 10 min at 12,000 g in 4 °C, with supernatant drawn. In total 200 µL of chloroform is mixed and subsequently shaken for 15 s, followed by 2 min of room temperature incubation. Samples are spun and supernatant is collected. A 1:1 mixture of the sample to 70% EtOH was put into a RNA purifying column (RNEasy Kit; Qiagen) and followed manufacturer instructions. To reverse transcribe to cDNA, RNA (2 ng µL⁻¹) was placed in final concentrations of 0.5 mM dNTPs (Invitrogen) and 50 ng µL⁻¹ random hexamers (Invitrogen). This solution was heated at 65 °C for 5 min and cooled to 4 °C. A mixture containing 50 units of SuperScript III (Invitrogen), 5× First-Strand Buffer (Invitrogen) and DTT (final concentration of 5 µM; Invitrogen) was added to the heated solution. The final mixture was held at room temperature (RT) for 5 min and subsequently, 50 °C for 1 h.

**Real-Time quantitative PCR (RT-qPCR).** Duplicate 25-ng cDNA samples were subjected to qPCR analysis using Rotor-Gene 6000 real-time rotary analyzer (Corbett Life Science; Concord NSW, Australia) with TaqMan Assay fluorogenic 5′nuclease chemistry (Invitrogen) as the fluorophore. Final concentrations for a 10-µL qPCR reaction with 25-ng of loaded cDNA include 0.25 U of AmpliTaq Gold DNA polymerase (Roche), 1.25 mM MgCl₂ (Roche), 100 µM dNTPs and 10× PCR buffer (Roche). Briefly, the samples are heated at 95 °C for 10 min. Samples are then subject to being heated at 95 °C for 10 s and then 58 °C for 45 s for a total of 45 cycles. Expression levels were normalized to that of peptidylprolyl isomerase A (PPIA) for tissues or Polr2A for cells mRNA using the delta–delta CT method (2⁻ᐃᐃCT). Gene expression data were gathered from various tissues. Arg 1 (Mm00475988_m1), Basph8 (Mm00484933_m1), Gata3 (Hs00231122_m1), Nos2 (Mm00440502_m1), Polr2a (Mm00839493_m1), Ppia (Mm02342430_g1), Siglecf (Mm00523987_m1), SERT (Mm00439391_m1), Tph1 spanning exons 2-3 (Mm01202614_m1), Tph1 probe spanning exon 4-5 (Mm00493794_m1), Tph2 (Mm00557715_m1), Tpsb2 (Mm01301240_g1), and Ucp1 (Mm01244861_m1) from Lifetechnologies were used.

**Liver triglycerides quantification.** Lipids were extracted from chipped liver (30–50 mg), homogenized in 1 mL of 2:1 chloroform: methanol, mixed overnight and subsequent isolation was carried out using an adapted Folch extraction protocol[57]. Briefly, samples are spun at 4500 g for 10 min at 4 °C, 200 µL of 0.9% NaCl is added and subsequently vortexed. Contents is spun again at the above conditions for 3 min. Clear bottom layer (400 µL) is drawn and resultant samples were freeze-dried and subsequently solubilized in 100% 2-propanol (200 µL). The Cayman Chemicals Triglyceride kit was used to assess triglyceride levels as directed by manufacturer instructions.

**Serotonin quantification.** Tissues were homogenized in 0.2 N perchloric acid in a tissue processing homogenizer (Percellys). The adipose tissue supernatant was extracted and mixed 1:1 in 1 M borate buffer (1 M boric acid, pH is brought up to 9.25 using concentrated NaOH). Platelet poor plasma is diluted 1:10. Tissues are undiluted. For mast cells, cells are spun down at 200 g for 5 min and media is extracted. Uninhibited and inhibited degranulated mast cell media are diluted 1:30 and 1:10, respectively. Serotonin concentrations are determined by an ELISA kit as per manufacturer's instructions (Serotonin EIA kit Beckman Coulter; IM1749). All aforementioned dilutions are done using the Dilution buffer provided by the Serotonin Kit.

**Flow Cytometry Analysis.** FCε-RI-PE (566608) and CD117-FITC (561680) antibodies (Becton Dickinson Canada Inc) were used (concentrations were optimized prior to experiment as per manufacturer recommendations) in combination with propidium iodide (Sigma-Aldrich) as a viability dye. Cultured mast cells were counted on a FACSCanto (BD Biosciences) and analyzed on FlowJo software 10.6.0 (FlowJo, LLC, Ashland, OR, USA). Briefly, 10⁶ cells per sample were placed in FACS tubes and washed with FACS buffer (Phospho-buffered saline with 5 mg/mL Bovine serum albumin). Samples are spun at 500 g and liquid decanted. After 30 min incubation with appropriate antibody in FACS buffer, the samples are spun, decanted and resuspended in FACS buffer. The resuspended sample is filtered through nylon mesh before running samples into FACS machine. Cells were gated on lymphocytes, single cells, live/dead or FCε-RI vs. CD117. Acid-killed cell sample was used to set the gates for the live/dead stain and unstained sample was used to set the gates for FCε-RI vs. CD117.

**Statistical analysis.** Data were evaluated by two-tailed Student's t-test or one-way ANOVA where appropriate. A repeated measures ANOVA was used for body weights and composition plots, thermography, GTT and ITT data. Fisher-LSD was used to evaluate significance between selected groups. Significance was accepted at $p < 0.05$ and data were presented as mean ± SEM. All measurements were taken from distinct samples collected from individual mice.

**Reporting summary.** Further information on research design is available in the Nature Research Reporting Summary linked to this article.

## Data availability

The BioGPS gene annotation portal using the "GeneAtlas MOE430, gcrma" dataset and 1419524_at Probeset (http://ds.biogps.org/?dataset=GSE10246&gene=21990) for the Tph1 gene was used with a correlation cut-off of 0.95 used in Fig. 1c. Relevant data supporting the findings of this study are readily available within supplementary information files and source data. Source data can be found for Figs. 1a, b, d–f, h, j, 2b–d, f–j, 3b–k, 4a–f, h, j, Supplementary Fig. 1a–g, Supplementary Fig. 2a–k, Supplementary Fig. 3a–l, Supplementary Fig. 4a–d, f, h. Source data are provided as a Source Data File. All data contained in this study are available upon reasonable request from the corresponding author.

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

## Acknowledgements

We thank Drs. Thomas Hawke and Jonathan Bramson from McMaster University for the use of microscope and flow cytometry equipment, respectively. We thank Dr. Gerard Karsenty from Columbia University for the generous gift of the Tph1 double floxed mice. This work was supported by grants from the Canadian Institutes of Health Research (201709FDN-CEBA-116200 to GRS and 144625-1 to GRS and WIK) and Diabetes Canada (DI-5-17-5302-GS to GRS). E.M.D. is a recipient of the Vanier Canada Graduate Scholarship. G.R.S. is supported by a Canada Research Chair and the J Bruce Duncan Endowed Chair in Metabolic Diseases.

## Author contributions

J.M.Y., E.M.D., K.M.M., J.D.C., W.I.K. and G.R.S. designed the experiments. J.M.Y., E.M.D., E.J.C., J.M.L., B.W., E.A.D., E.D.C., W.W., and J.D.C. conducted the experiments. J.M.Y. and G.R.S. wrote the paper. Diagrams and drawn pictures on Figs. 1–3 were created by J.M.Y. on PowerPoint Office 365 and Procreate v4.3.9 for Mac iOS iPad Pro, respectively. All authors offered edits to final manuscript.

## Competing interests

J.M.Y., E.M.D., E.J.C., J.M.L., B.W., E.A.D., E.D.C., W.W., J.D.C., W.I.K. declare no competing interests. K.M.M. provides advisory services to Akcea Therapeutics Canada Inc. G.R.S. has received honoraria/consulting fees from Astra Zeneca, Boehringer, Eli-Lilly, Esperion Therapeutics, Novo Nordisk, Poxel, Pfizer, Merck, Rigel Therapeutics and Terns Therapeutics. G.R.S. has received research funding from Esperion Therapeutics. G.R.S. and W.I.K. hold a patent for inhibiting peripheral serotonin for the treatment of metabolic diseases including obesity.
