## [Peer Review File · Nature Communications]

Reviewers' comments:

Reviewer #1 (Remarks to the Author):

In this elegant study, it is shown that mast cell-derived 5HT inhibits adipocyte thermogenesis in specific adipose tissue depots.

The data are convincing, and the manuscript is of wide appeal.

However, there needs to be a discussion of how this model fits with clear evidence (from Morrison lab and others) that show central 5-HT, as well as 5-HT evoked by the psychostimulant MDMA increase energy expenditure and BAT and iWAT activity.

There should also be some discussion of the potential effects of 5-HT on vascular tone and heat loss.

Reviewer #2 (Remarks to the Author):

This is a potentially important study, in that it reopens the question of the role of mast cells and their products (in this case, serotonin) in obesity and insulin resistance. The study will most likely be compared to that produced by the Rodewald group and published in *Cell Metabolism* (Gutierrez et al. *Cell Metabolism* 21, 678-691, 2015) entitled: "Hematopoietic Kit deficiency, rather than lack of mast cells, protects mice from obesity and insulin resistance".

In that study, the authors provided evidence that hematopoietic effects of the mutations in c-kit other than the mast cell-deficiency of the mice were responsible for protecting the mice from the development of diet-induced obesity and insulin resistance. By contrast, in the present study, the authors present evidence, some of it derived from studies of mast cell-engrafted Kit W-shW-sh mice, indicating that mast cell serotonin synthesis can help to prevent the development of obesity and insulin resistance.

Can both studies be correct? In my judgement, yes, but I think that the authors really should make more of a point in contrasting the important details pertaining to each study. They should also deal

more explicitly with differences in results obtained with the mast cell-engrafted Kit W-shW-sh mice and those obtained with the mast cell-deficient Cpa3-Cre mice

1). In the case of the Gutierrez study, the mice were apparently kept at “room temperature” (presumably, ~20 degrees C, although I couldn’t find that statement in the paper). In the present study, I believe that the key experiments were done with the mice maintained in a “thermoneutral” temperature. (This is defined on line 58 as “30 degrees C” and in the Methods (line 261) as “28 degrees C”. Presumably, one of these needs to be corrected.).

This key difference should be made more explicit, by dealing with it in more detail in the Discussion. Also, it should be made easier for the reader to see which experiments were conducted at “thermoneutrality”. All of the figures, not just Fig. 1, should include in the title “thermoneutrality” when that condition was examined, so that the reader keeps this important point in mind when reading the results.

I think that the title of the manuscript should also emphasize the difference between this study and the earlier, Gutierrez et al., report. A possible title would be: “Genetic deletion of mast cell serotonin synthesis prevents the development of obesity and insulin resistance at thermoneutrality”. In the Abstract (line 31) the authors should define “thermoneutrality” by adding “....to thermoneutrality (28 or 30 degrees C) promotes the.....”.

2). Some of the results obtained with mast cell-engrafted Kit W-shW-sh mice and those obtained with the mast cell-deficient Cpa3-Cre mice are quite different. More should be said about such differences. For example, effects shown in fig. 3 E and F (with the Cpa3-Cre mice) are much more modest than those shown for the mast cell-engrafted Kit W-shW-sh mice in fig. 2 I and J. Similarly, fig. 3 should include results paralleling those shown for frequency of adipocyte size in fig. 2 G and H. The authors should acknowledge the possibility that some of these differences may reflect other problems with the Kit W-shW-sh mice than their mast cell deficiency.

3). The authors have transferred in vitro derived BMMC into Kit W-shW-sh mice, but they don’t give mast cell counts in various tissues, except for mast cells in iWAT and gWAT (suppl. fig. 2 F and G). Mast cells were transferred iv, into the skin and into the peritoneal cavity, and counts for the numbers of mast cells in these sites also should be given. The point is to try to establish whether or not the lack of Tph1 interfered with the transfer/survival of mast cells in various tissues, such as the skin or peritoneal cavity, GI tract, etc.

4). More should be said about the disparate results obtained regarding mast cells and obesity than “However, whether there is a causal relationship between MCs and obesity is equivocal (25, 26).” As noted above, the room temperatures of the prior studies could be noted. However, ref. 26 clearly showed that, in the model of obesity examined, it was not the mast cell-deficiency of the Kit W/W-v mice but some other effect(s) of the c-kit mutations that was responsible. Furthermore, lines 95-96

could be altered to say: “We examined whether MC-derived serotonin can contribute to obesity and insulin resistance at thermoneutrality, which wasn’t addressed in either of the prior two studies (25, 26).”

Minor comments.

1). In lines 111 and 112, figures 2 I and 2 J should be cited instead of 2 G and 2 H.

2). Line 160 should begin with “multilocular” instead of “multiocular”.

3). In the Methods, the authors should state for how long each of the different types of mice studied were kept at their animal facility.

4). Line 239, ref. 40 does not report the Cpa3-Cre mice (should be ref. 36).

5). In line 263, what do the authors mean by the mice being randomized and “weight matched if possible”?

Response to Reviewers

Reviewer #1 (Remarks to the Author):

In this elegant study, it is shown that mast cell-derived 5HT inhibits adipocyte thermogenesis in specific adipose tissue depots.

The data are convincing, and the manuscript is of wide appeal.

We thank the reviewers for their positive comments and have inserted the following paragraphs into the discussion of the paper as suggested.

However, there needs to be a discussion of how this model fits with clear evidence (from Morrison lab and others) that show central 5-HT, as well as 5-HT evoked by the psychostimulant MDMA increase energy expenditure and BAT and iWAT activity.

We have inserted the following paragraph (Lines 208-223):

In the current study, the increase in energy expenditure that we¹³ and others¹⁴ have observed with whole-body inhibition of Tph1 and subsequent reductions in circulating serotonin can be recapitulated by removing Tph1 in mast cells. These findings seem paradoxical based on previous findings that Tph2⁺ serotonergic neurons in the brain are linked to BAT³⁰ and the central injection of serotonin³¹ enhances BAT energy expenditure while serotonin receptor antagonists³² or the inducible deletion of serotonergic neurons reduce BAT activity and iWAT Ucp1 when mice are housed at room temperature³³. These opposing functions of central and peripheral serotonin are consistent with findings from other highly conserved regulators of energy balance such as the AMP-activated protein kinase where genetic reductions of hypothalamic AMPK increases energy expenditure³⁴, while reductions of AMPK in adipose tissue lower energy expenditure and iWAT browning³⁵. A similar paradigm of opposing functions between central and adipose specific mTOR has also been observed³⁶⁻³⁸. Thus, there is a precedent by which central and peripheral pathways may exert opposing functions on energy expenditure. Collectively, these data support a model where Tph2 and central serotonin enhance energy expenditure in response to cold, but under thermoneutral conditions or with obesity, peripheral serotonin synthesis by mast cell Tph1 reduces thermogenic activation, thus lowering adipose tissue energy expenditure.

There should also be some discussion of the potential effects of 5-HT on vascular tone and heat loss.

We have inserted the following paragraph in to the discussion (Lines 279-291):

Another important consideration when interpreting our findings on energy expenditure are that serotonin is a potent regulator of vascular tone (reviewed here⁴⁷). The regulation of vasculature tone is extremely complex and depending on the tissues and serotonin receptors expressed, serotonin can induce either vasodilation or vasoconstriction to regulate heat loss and heat retention, respectively (reviewed here⁴⁷). Mice with germline deletions of Tph1 have altered blood pressure and vascular tone⁴⁸ and chemical inhibition of Tph1 can reduce hypoxia-induced pulmonary-arterial hypertension⁴⁹. However, we have shown that in obese mice fed a HFD,

similar to the current study, that treatment with the pan-Tph inhibitor, LP533401, does not affect heart rate or blood pressure, and that the effects of this compound on weight loss and insulin sensitivity are completely dependent on the expression of Ucp1¹³. These data suggest that in the context of high-fat diet-induced obesity it is unlikely that genetic deletion of Tph1 specifically in mast cells, which importantly had no effect on circulating serotonin, would elicit any effect on vasculature tone or heat loss. However, future studies directly investigating this possibility are warranted.

Reviewer #2 (Remarks to the Author):

This is a potentially important study, in that it reopens the question of the role of mast cells and their products (in this case, serotonin) in obesity and insulin resistance. The study will most likely be compared to that produced by the Rodewald group and published in Cell Metabolism (Gutierrez et al. Cell Metabolism 21, 678-691, 2015) entitled: “Hematopoietic Kit deficiency, rather than lack of mast cells, protects mice from obesity and insulin resistance”.

In that study, the authors provided evidence that hematopoietic effects of the mutations in c-kit other than the mast cell-deficiency of the mice were responsible for protecting the mice from the development of diet-induced obesity and insulin resistance. By contrast, in the present study, the authors present evidence, some of it derived from studies of mast cell-engrafted Kit W-shW-sh mice, indicating that mast cell serotonin synthesis can help to prevent the development of obesity and insulin resistance.

Request A: Can both studies be correct? In my judgement, yes, but I think that the authors really should make more of a point in contrasting the important details pertaining to each study.

We have inserted the following paragraph into the discussion (Lines 238-264):

Our findings also raise important questions about the role of mast cells in obesity and insulin resistance and the potential limitation of mouse models of mast cell deficiency to establish causality. Recent studies by Gutierrez and colleagues²⁸ indicated that previous findings linking mast cells with obesity-induced insulin resistance²⁷ may have been an artifact of the Kit mutation and not a direct result of a reduction in mast cells. These conclusions were based on the findings that: 1) the restoration of mast cells in Kit^{W/W^v} through Kit^{+/+} hematopoietic reconstitution leads to weight gain when mice are fed a HFD and 2) mice containing an overexpressing Cre recombinase linked to Cpa3, which results in the death of mast cells, are not protected from HFD-induced obesity or insulin resistance. It should be noted that our studies differ in many ways from those of Gutierrez²⁸ or Liu and colleagues²⁷ as we did not directly test the role of mast cells per se in the regulation of obesity, but rather made comparisons between mast cells with or without Tph1 using two unique and completely distinct mouse models. First, we showed that reconstitution of Kit^{W-sh/W-sh} mice with Tph1^{+/+} mast cells increased weight gain and insulin resistance, while Tph1^{-/-} mast cells did not. It is important to note this difference in weight gain between Kit^{Tph1+/+} and Kit^{Tph1-/-} mice was not due to differences in the number of mast cells within WAT, which were comparable upon reconstitution. Secondly and perhaps most importantly, given the potential confounding effects of the Kit inversion mutation on metabolism²⁸, we generated an entirely distinct mouse model in which Tph1 floxed mice were crossed to Cpa3^{Cre/+} mice — the preferential model suggested by Gutierrez and colleagues²⁸ for metabolic studies since it does not induce a genotoxic effect on mast cells — and observed

similar results with respect to weight gain, adipose tissue Ucp1, energy expenditure and insulin sensitivity in mice lacking mast cell Tph1. And while it is known that Cpa3 is also expressed in basophils⁴², it is important to note that mast cells have a much higher expression of Tph1 (>1000-fold) compared to basophils or other immune cells types such as macrophages making it unlikely that these cell types contributed to the phenotype⁴³. Therefore, our findings, in two distinct mouse models as suggested^{28,44} support the conclusions that mast cell Tph1 protects against obesity induced insulin resistance by enhancing adipose tissue Ucp1 and whole body energy expenditure.

Request B: They should also deal more explicitly with differences in results obtained with the mast cell-engrafted Kit W-shW-sh mice and those obtained with the mast cell-deficient Cpa3-Cre mice.

1). In the case of the Gutierrez study, the mice were apparently kept at “room temperature” (presumably, ~20 degrees C, although I couldn’t find that statement in the paper). In the present study, I believe that the key experiments were done with the mice maintained in a “thermoneutral” temperature. (This is defined on line 58 as “30 degrees C” and in the Methods (line 261) as “28 degrees C”. Presumably, one of these needs to be corrected.) This key difference should be made more explicit, by dealing with it in more detail in the Discussion. Also, it should be made easier for the reader to see which experiments were conducted at “thermoneutrality”. All of the figures, not just Fig. 1, should include in the title “thermoneutrality” when that condition was examined, so that the reader keeps this important point in mind when reading the results.

Only the experiment in Figure 1 was conducted at both room temperature and thermoneutrality. The remaining experiments were conducted at room temperature and we have now stated this more clearly in the methods as indicated below (Lines 324-328):

Mice were maintained on a 12-hour light and dark cycle (lights on at 0700) with food and water provided ad libitum. For experiments conducted in Figure 1, mice were housed at either room temperature (23 °C) or thermoneutrality (29 °C) and were fed a 60% HFD starting at 8 weeks of age for 10 weeks. All other experiments were performed in mice housed solely at room temperature.

I think that the title of the manuscript should also emphasize the difference between this study and the earlier, Gutierrez et al., report. A possible title would be: “Genetic deletion of mast cell serotonin synthesis prevents the development of obesity and insulin resistance at thermoneutrality”. In the Abstract (line 31) the authors should define “thermoneutrality” by adding “....to thermoneutrality (28 or 30 degrees C) promotes the.....”.

As stated above, as all of the genetic experiments in which we demonstrated a causal link between mast cell Tph1 were done under “standard” room temperature conditions. Therefore we believe the current title is appropriate.

2). Some of the results obtained with mast cell-engrafted Kit W-shW-sh mice and those obtained with the mast cell-deficient Cpa3-Cre mice are quite different. More should be said about such

differences. For example, effects shown in fig. 3 E and F (with the Cpa3-Cre mice) are much more modest than those shown for the mast cell-engrafted Kit W-shW-sh mice in fig. 2 I and J. Similarly, fig. 3 should include results paralleling those shown for frequency of adipocyte size in fig. 2 G and H. The authors should acknowledge the possibility that some of these differences may reflect other problems with the Kit W-shW-sh mice than their mast cell deficiency.

We have inserted the following paragraph (Lines 265-278):

While these models are complementary, there are important differences in our results. The first and most evident is that the protection from obesity-induced glucose intolerance and insulin resistance is more dramatic in the Kit^{Tph1^{-/-}} mice (Figure 2) compared to the Tph1 MCKO mice (Figure 3). This difference is most likely attributed to sex differences between models; since males were studied in the Kit mast cell reconstitution experiments while female mice were studied in the Tph1 floxed experiments. As female mice are lighter and more insulin sensitive than male mice at the same age and duration of high-fat diet feeding⁴⁵, this likely underlies the primary difference between the models. Other reasons for the more modest phenotype in the Tph1 MCKO mice is the time of testing which was after 8 weeks of high-fat diet compared to the 12 weeks for the Kit^{Tph1^{-/-}} mice. The longer duration of high-fat diet feeding in the Kit^{Tph1^{-/-}} mice was utilized to maximize mast cell engraftment as previously suggested⁴⁶. Therefore, sex and duration of the high-fat diet before testing may explain the differences between models. However, we believe showing a similar phenotype in two difference sexes, albeit to a lesser degree in the female mice, is an important strength of the study.

Frequency of adipocyte size of the Tph1 MCKO mice can be observed in the H&E stain in Figure 4G in iWAT and Figure 4I in gWAT. The representative histological images are quite drastic since the frequency of smaller adipocytes is clear and we do not believe a frequency plot is required. This analysis was carried out in the Kit adoptive experiments since the effects were quite modest in comparison to the Tph1 MCKO mice.

3). The authors have transferred in vitro derived BMMC into Kit W-shW-sh mice, but they don't give mast cell counts in various tissues, except for mast cells in iWAT and gWAT (suppl. fig. 2 F and G). Mast cells were transferred iv, into the skin and into the peritoneal cavity, and counts for the numbers of mast cells in these sites also should be given. The point is to try to establish whether or not the lack of Tph1 interfered with the transfer/survival of mast cells in various tissues, such as the skin or peritoneal cavity, GI tract, etc.

Unfortunately we do not have any tissue remaining from this experiment and thus cannot measure the transfer/survival of mast cells into any other tissues than those presented in the manuscript where we showed that the number of mast cells in iWAT and gWAT were comparable between genotypes. As iWAT is a subcutaneous adipose tissue depot that resides adjacent to the skin and gWAT is a visceral adipose tissue depot that resides in the intraperitoneal cavity we believe these tissues are appropriate surrogates for the requested experiment.

4). More should be said about the disparate results obtained regarding mast cells and obesity than "However, whether there is a causal relationship between MCs and obesity is equivocal (25, 26)." As noted above, the room temperatures of the prior studies could be noted. However, ref.

26 clearly showed that, in the model of obesity examined, it was not the mast cell-deficiency of the Kit W/W-v mice but some other effect(s) of the c-kit mutations that was responsible. Furthermore, lines 95-96 could be altered to say: “We examined whether MC-derived serotonin can contribute to obesity and insulin resistance at thermoneutrality, which wasn’t addressed in either of the prior two studies (25, 26).”

We have removed our simple statement from the results section about the findings being equivocal and as noted in Request A have now provided a very detailed paragraph in the discussion about how our current studies differ from the findings of Gutierrez and colleagues.

Minor comments.

1). In lines 111 and 112, figures 2 I and 2 J should be cited instead of 2 G and 2 H.

This has been corrected.

2). Line 160 should begin with “multilocular” instead of “multiocular”.

This has been corrected.

3). In the Methods, the authors should state for how long each of the different types of mice studied were kept at their animal facility.

This has been added.

4). Line 239, ref. 40 does not report the Cpa3-Cre mice (should be ref. 36).

This has been corrected.

5). In line 263, what do the authors mean by the mice being randomized and “weight matched if possible”?

We have clarified what was meant by this comment in the methods by inserting the following sentences (Lines 328-334):

For bone marrow-derived MC reconstitution experiments (related to Figures 2 and 4) Kit^{W-sh/W-sh} mice⁷ at 9-10 weeks of age were separated into three different weight-matched groups and were injected with a total of 20 million MCs from Tph1^{+/+} (Kit^{Tph1+/+}) or Tph1^{-/-} (Kit^{Tph1-/-}) mice at three different anatomical locations (10 million intravenously through tail vein, 5 million subcutaneously (dorsal interscapular area) and 5 million intraperitoneally). The third group of Kit^{W-sh/W-sh} mice received an injection of equal volume saline at all three sites (Kit^{sham}). C57Bl/6J mice were also injected with equivalent amounts of saline at all aforementioned sites.

REVIEWERS' COMMENTS:

Reviewer #1 (Remarks to the Author):

My concerns are addressed, I thank the authors for their detailed responses.

Reviewer #2 (Remarks to the Author):

The authors have attempted to respond to my questions, and the manuscript's message has been enhanced. I have a few additional points for them to consider.

1. The abnormalities related to mast cells in the *Kit^{W-sh/W-sh}* mice are not due to mutations in *Kit* per se, so I think that a few parts of the manuscript need to be revised to accommodate this.

For example, in the paragraph beginning on line 142, I suggest changing the text to: "To further substantiate....metabolic effects of the mutation in the *Kit^{W-sh/W-sh}* mice in other tissues,....".

In lines 244-245, it would be more accurate to say: "...may have been due to effects of the *Kit^{W-sh/W-sh}* mutation other than a direct result of the animal's reduction in mast cells."

2. The description of the findings obtained by Gutierrez et al. (lines 245-249) is not fully accurate. It would be better to say: "1) the restoration of hematopoietic cells in *Kit^{W/W-v}* mice, whether or not the hematopoietic cells could generate mast cells, lead to weight gain when mice were fed a HFD...".

3. Similarly, the statement in lines 258-260 needs revision: "..., we generated an entirely distinct mouse model in which Tph1 floxed mice were crossed to *Cpa3^{Cre/+}* mice – the preferential model suggested by Gutierrez and colleagues²⁸ for metabolic studies since it does not induce a genotoxic effect on mast cells –".

According to the Methods section (lines 307-308), the *Cpa3^{Cre/+}* mice used in the present study were not the same as the mice used by Gutierrez and colleagues²⁸. It would be more accurate to say simply: "..., we generated an entirely distinct mouse model in which Tph1 floxed mice were crossed to *Cpa3^{Cre/+}* mice for metabolic studies since this does not induce a genotoxic effect on mast cells –".

4. The authors may be aware that there has been some dispute about whether or not human mast cells can contain serotonin. Perhaps the most definitive study showed that human mast cells can contain some serotonin, but (at least for the mast cells analyzed) significantly less than that in the tested mouse mast cells (J Allergy Clin Immunol. 2007 Feb;119(2):498-9. Epub 2006 Oct 13. Human mast cells are capable of serotonin synthesis and release. Kushnir-Sukhov NM, Brown JM, Wu Y, Kirshenbaum A, Metcalfe DD). It would be good to include this reference and for the authors to comment on the matter of human mast cells as a source of 5-HT, perhaps at the end of the discussion.

5. Typos:

Line 281: it should be: "...a similar phenotype in two different sexes, albeit..."

Fig. 2A: it should be: "MC 5-HT" not just "5-HT".

Legend for Figure S2: in the fourth line, it should be "(D)" (not "(A)"), and then "(D)-(F)" becomes "(E)-(G)". The symbol "(H)" should be introduced before "Tph1 expression..." then "(G)" should become "(I)" and "(H)" should become "(J)".

Point-By-Point Responses to Referees

To the Reviewers of our manuscript entitled “*Genetic Deletion of Mast Cell Serotonin Synthesis Prevents the Development of Obesity and Insulin Resistance*” by Yabut et al., we have looked at your comments and concerns and have promptly addressed them below:

**Please be warned that the comment lines may have shifted if tracked comments/edits have been removed from the document.*

REVIEWERS' COMMENTS:

Reviewer #1 (Remarks to the Author):

My concerns are addressed, I thank the authors for their detailed responses.

- We thank Reviewer #1 for their contributions to the revision of our paper.

Reviewer #2 (Remarks to the Author):

The authors have attempted to respond to my questions, and the manuscript’s message has been enhanced. I have a few additional points for them to consider.

1. The abnormalities related to mast cells in the $Kit^{W-sh/W-sh}$ mice are not due to mutations in Kit per se, so I think that a few parts of the manuscript need to be revised to accommodate this.

For example, in the paragraph beginning on line 142, I suggest changing the text to:

“To further substantiate....metabolic effects of the mutation in the $Kit^{W-sh/W-sh}$ mice in other tissues,....”.

- We have since changed this on lines 156-157, “*To further substantiate the observations that mast cell $Tph1$ promotes obesity and insulin resistance without the potential confounding metabolic effects of the $Kit^{W-sh/W-sh}$ mutation*”

In lines 244-245, it would be more accurate to say: “...may have been due to effects of the $Kit^{W-sh/W-sh}$ mutation other than a direct result of the animal’s reduction in mast cells.”

- We have since changed this on lines 268-269, “*may have been due to effects intrinsic to the $Kit^{W-sh/W-sh}$ mutation rather than a direct result of mast cell reduction*”.

- Another instance of this “*Kit inversion*” occurs on line 289, and has been since changed to $Kit^{W-sh/W-sh}$ mutation.

2. The description of the findings obtained by Gutierrez et al. (lines 245-249) is not fully accurate. It would be better to say: “1) the restoration of hematopoietic cells in $Kit^{W/W-v}$ mice, whether or not the hematopoietic cells could generate mast cells, lead to weight gain when mice were fed a HFD...”.

- We have since changed this on lines 270-279, “*1) the restoration of hematopoietic cells in $Kit^{W/Wv}$ mice, whether or not the hematopoietic cells could generate mast cells, led to weight gain when mice are fed a HFD*”.

3. Similarly, the statement in lines 258-260 needs revision: “... we generated an entirely distinct mouse model in which $Tph1$ floxed mice were crossed to $Cpa3^{Cre/+}$ mice – the preferential model suggested by Gutierrez and colleagues²⁸ for metabolic studies since it does not induce a genotoxic effect on mast cells – “.

Point-By-Point Responses to Referees

According to the Methods section (lines 307-308), the Cpa3^{Cre/+} mice used in the present study were not the same as the mice used by Gutierrez and colleagues²⁸. It would be more accurate to say simply: "...we generated an entirely distinct mouse model in which Tph1 floxed mice were crossed to Cpa3^{Cre/+} mice for metabolic studies since this does not induce a genotoxic effect on mast cells – “.

- We have simply stated on lines 290-291, *“we generated an entirely distinct mouse model in which Tph1 floxed mice were crossed to Cpa3^{Cre/+} mice”* and removed the above comment.

4. The authors may be aware that there has been some dispute about whether or not human mast cells can contain serotonin. Perhaps the most definitive study showed that human mast cells can contain some serotonin, but (at least for the mast cells analyzed) significantly less than that in the tested mouse mast cells (J Allergy Clin Immunol. 2007 Feb;119(2):498-9. Epub 2006 Oct 13. Human mast cells are capable of serotonin synthesis and release. Kushnir-Sukhov NM, Brown JM, Wu Y, Kirshenbaum A, Metcalfe DD). It would be good to include this reference and for the authors to comment on the matter of human mast cells as a source of 5-HT, perhaps at the end of the discussion.

- We have added this reference and comment on human mast cells serotonin on lines 336-339, *“Lastly, while human mast cells synthesize serotonin via Tph1⁵¹, it is important to highlight that there are relatively small amounts of serotonin secreted and stored in human mast cells in comparison to rodents. Future studies investigating the role of mast cell serotonin synthesis in humans and the potential impact this may have on vasculature tone are warranted.”*

5. Typos:

Line 281: it should be: "...a similar phenotype in two **different** sexes, albeit..."

- This has been corrected on line 319.

Fig. 2A: it should be: *“MC 5-HT”* not just *“5-HT”*.

- This has been corrected in Figure 2A on the diagram.

Legend for Figure S2: in the fourth line, it should be *“(D)”* (not *“(A)”*), and then *“(D)-(F)”* becomes *“(E)-(G)”*. The symbol *“(H)”* should be introduced before *“Tph1 expression...”* then *“(G)”* should become *“(I)”* and *“(H)”* should become *“(J)”*.

- This has since been corrected. All figures and supplemental figures were looked after to ensure consistency.